# Characterisation of Novel Angiogenic and Potent Anti-Inflammatory Effects of Micro-Fragmented Adipose Tissue

**DOI:** 10.3390/ijms22063271

**Published:** 2021-03-23

**Authors:** Baoqiang Guo, Xenia Sawkulycz, Nima Heidari, Ralph Rogers, Donghui Liu, Mark Slevin

**Affiliations:** 1Manchester Metropolitan University, Chester Street, Manchester M1 5GD, UK; B.guo@mmu.ac.uk (B.G.); XENIA.SAWKULYCZ@stu.mmu.ac.uk (X.S.); n.heidari@gmail.com (N.H.); prcldh@hotmail.com (D.L.); 2The Regenerative Clinic, Harley Street, London W1G 6JP, UK; 3Next AI, Leadenhall Street, London EC3V 1LP, UK; 4Rogers Regenerative Medical Group, Harley Street, London W1U 2HX, UK; dr@ralphrogers.com; 5University of Medicine and Pharmacy, Science and Technology, W1G 7ET Târgu Mures, Romania

**Keywords:** micro-fragmented adipose tissue, mesenchymal stem cells, tissue regeneration, angiogenesis, inflammation

## Abstract

Adipose tissue and more specifically micro-fragmented adipose tissue (MFAT) obtained from liposuction has recently been shown to possess interesting medicinal properties whereby its application supports pain reduction and may enhance tissue regeneration particularly in osteoarthritis. Here we have characterised samples of MFAT produced using the Lipogems^®^ International Spa system from eight volunteer individuals in order to understand the critical biological mechanisms through which they act. A variation was found in the MFAT cluster size between individual samples and this translated into a similar variation in the ability of purified mesenchymal stem cells (MSCs) to form colony-forming units. Almost all of the isolated cells were CD105/CD90/CD45+ indicating stemness. An analysis of the secretions of cytokines from MFAT samples in a culture using targeted arrays and an enzyme-linked immunosorbent assay (ELISA) showed a long-term specific and significant expression of proteins associated with anti-inflammation (e.g., interleukin-1 receptor alpha (Il-1Rα) antagonist), pro-regeneration (e.g., hepatocyte growth factor), anti-scarring and pro-angiogenesis (e.g., transforming growth factor beta 1 and 2 (TGFβ1/2) and anti-bacterial (e.g., chemokine C-X-C motif ligand-9 (CXCL-9). Angiogenesis and angiogenic signalling were notably increased in primary bovine aortic endothelial cells (BAEC) to a different extent in each individual sample of the conditioned medium whilst a direct capacity of the conditioned medium to block inflammation induced by lipopolysaccharides was shown. This work characterises the biological mechanisms through which a strong, long-lasting, and potentially beneficial effect can be observed regarding pain reduction, protection and regeneration in osteoarthritic joints treated with MFAT.

## 1. Introduction

It has been known for decades that fat tissue has extraordinary capabilities and a broad range of physiological functions [1] and more recently the impressive impact of the use of micro-fragmented adipose tissue (MFAT) and its cellular constituents has been documented clinically in effective treatments for osteoarthritis, post-menopausal vaginal atrophy, the repair of perianal fistulae and diabetic foot [2,3,4,5].

In order to maximise the pain relief and protective and regenerative capacity of MFAT in a disease, it is important to fully understand the biological mechanisms through which it acts. Previous work has shown that mesenchymal stem cells (MSCs) derived from the microvessel pericytes from MFAT are critical to the functionality and possess strong anti-inflammatory properties and, in addition, secrete factors over a considerable length of time (when delivered in the form of a tissue graft) that could help to encourage natural tissue regeneration and repair [6,7].

Recent studies using animal models/ex vivo cultures have demonstrated the capacity of MFAT to reduce inflammatory sepsis in a mouse caecal ligation model effectively reducing mortality [8] and increase human tendon stem cell proliferation associated with enhanced vascular endothelial growth factor (VEGF) expression indicating a role for stimulating both cartilage regeneration and increased vascularisation [9].

In this work, for the first time we have fully characterised and compared MFAT samples from eight individual donors obtained from liposuction and processed by a Lipogems^®^ International Spa processing device. We indicate the morphological characteristics, variations and critical cytokine outputs in relation to inflammation, anti-bacterial capability, regenerative ability and angiogenesis/tissue repair.

## 2. Materials and Methods

### 2.1. Preparation and Harvesting of MFAT

MFAT was obtained and processed from volunteers’ adipose tissue according to the policies of Manchester Metropolitan University. This was reviewed by the local board for human studies local ethical committees (EthOS; project ID 24407). All tissue samples were obtained after signed informed consent (five males and three females aged from 25–58 with no underlying medical conditions). Subcutaneous fat was obtained from abdominoplasty samples. MFAT was prepared as previously described [10]. Briefly, around 100 mL of adipose tissue was processed using an MFAT device (provided free of charge for this study by Lipogems^®^ International Spa, Milan, Italy). The lipoaspirate was then injected into the MFAT processing kit, which was a disposable and closed device filled with a saline solution that reduced the size of the adipose tissue clusters by means of an “enzyme free” minimal manipulation in a closed and aseptic system with two different sized filters. Experiments were performed on a minimum of five and a maximum of all eight of the samples depending on the availability and quantity of MFAT obtained from each individual.

### 2.2. Quantification of MFAT

In the current experiments, we first needed to ensure the consistency of the cellular material in each sample (as this can be variable from syringe to syringe). Hence, we established a simple method to quantify the amounts of MFAT by centrifuging (300 g/5 min) MFAT in a phosphate buffered saline (PBS) solution after 3 × washing (removing the interface). We used cut-blue-tips to obtain a 0.25 mL sample and characterised this as one unit. The counting of clusters of MFAT attached to the flask was performed after 24 h culture in T25 flasks with ethanol fixation and Giemsa staining whereby 10 × 40 fields from each flask (sample) were measured and the results displayed as a mean ± SD for each 0.25 mL sample.

### 2.3. Isolation, Expansion and Characterisation of Adipose-Derived (AD)-MSCs

Based upon the protocols of Agostini et al. [11], human MFAT samples (2 mL for each sample) were processed for MSC isolation. MFAT samples were washed three times with PBS for blood residual removal by centrifugation at 250× *g* for 3 min. After the liquid phase removal, MFAT samples were digested in a collagenase type I (1 mg/mL) solution (Thermo Fisher Scientific (TFS), Altrincham, UK) under gentle agitation for 1 h at 37 °C followed by neutralisation of the collagenase I with a minimal essential medium (MEM)α containing 20% foetal calf serum (FCS) and centrifuging at 800 g for 5 min to separate the stromal cell fraction (SVF) pellet from the adipocytes. The adipose stem cell (ASC) fraction was treated with a red blood cell lysis buffer (160 mM) of NH_4_Cl for 10 min on ice and then centrifuged at 800 g for 5 min. The supernatant was discarded, and the cell pellet was re-suspended in 3 mL of MEMα containing 20% FCS. After filtration with a 100 µm nylon strainer, which was washed with an additional 3 mL of the medium, the total 6 mL of cell suspension was pelleted and cultured in MEMα containing 20% FCS. The following day, unattached cells were washed away with PBS followed by the addition of a new MEMα medium containing 20% FCS. The change of medium was done every two to three days.

### 2.4. Flow Cytometric Based Phenotype Characterisation of the AD-MSCs Derived from the Lipogems^®^ International Spa System

AD-MSCs (100 µL) were labelled for 20 to 30 min on ice with manufacturer recommended concentrations of the mouse anti-human monoclonal fluorescent antibodies for CD73, CD90, CD105, CD45 or the isotype control (EBioscience). After harvesting, the cells were washed with PBS without magnesium and calcium. Cells were centrifuged sufficiently so that the supernatant fluid was removed with a minimal loss of cells but not so severe that the cells were difficult to resuspend. Cells were washed by centrifugation at 400 g for 5 min. the cell suspension was adjusted to a concentration of 1–2.5 × 106 cells/mL in ice-cold PBS, 0.1% bovine serum albumin (BSA) and 0.04% sodium azide. Cells were stained in polystyrene round-bottom fluorescence activated cell sorter (FACS) Falcon tubes. A total of 0.25–5 µL of conjugated primary antibody (EBioscience of TFS) was added into the 0.1 µL cell suspension. The cells were incubated with the conjugated antibody for at least 30 min in the fridge at 4 °C and kept in the dark. Cells were washed three times by centrifugation at 400 g for 5 min and resuspended in 500 μL to 1 mL of ice-cold PBS, 0.1% BSA and 0.04% sodium azide. The cells were kept in the dark on ice or at 4 °C in a fridge until the scheduled time for analysis. For the best results, our cells were always analysed on the flow cytometer straight after we finished the washing on a FACS-Calibur instrument (Becton Dickinson, Berkshire, UK).

In this case, only one sample of AD-MSCs was assessed and hence a statistical analysis or a variation from sample to sample were not able to be considered.

### 2.5. Quantification of Effective MSCs by Measurement of CFU-F Derived from the SVF of MFAT

The fibroblastoid colony-forming unit (CFU-F) assay is the standard to define the number of progenitor cells. This assay, modified from the one utilised for bone marrow MSCs [12], required 11–14 days of culture. The cell suspension of SVF from six of the eight samples was seeded at a low density (40–400 cells/cm^2^) to allow each clone to grow separately in a medium carefully chosen to allow clone growth [13]. After fixation and staining, clones with >50 cells were counted. In general, the frequency of stromal progenitors ranged from 1% to 10% relative to the total nucleated cell population. Each CFU-F assay was performed with two or more cell concentrations in triplicate for each donor to minimise assay variations and a representative example is shown here where colonies were counted from 10 fields of view for each sample at × 40. The number of colonies allowed for an estimation of the rate of doubling of the population during the primary phase of culture. The information gained from the CFU-F was particularly useful to enhance the quality control of any resulting cell therapeutic product.

### 2.6. Measurement of MFAT Critical Cytokine Expression

A Quantikine multiplex system (R&D Systems Inc, Oxford, UK) was used containing 32 self-designed dedicated proteins of specific relevance to regenerative and reparative medicine (see Appendix A). In addition, a stand-alone ELISA assay kit was used to analyse the expression of TGF-β1-3 purchased from Invitrogen (Warrington, UK) as this was not available on the array but is a critical cytokine group involved in scarless wound healing and angiogenesis.

Fifty µl of a conditioned culture medium (from MFAT samples of seven of the eight volunteers cultured in T25 cm flasks) or a control medium obtained after 24 h of culture and five days of culture in a serum free medium (MEMα) were applied to the array. The array was processed by the R&D Systems quality control technical team and the raw data were provided as a heat map and Excel charts as shown in Appendix A. The ELISA used standard sandwich protocols as described on their website. The cytokines showing a consistently increased expression are described in the results. 

### 2.7. Angiogenesis Assay BAEC Cell Tube Formation

Bovine aortic endothelial cells (BAEC) at passage 8–10 were maintained in an EBM2 medium supplemented with 2% FBS and growth factors as recommended by the manufacturer under a humidified air of 95% with 5% CO_2_ at 37 °C in a T-75 flask. The subconfluence cells were serum starved by replacing the growth medium with an endothelial cell growth basal medium (EBM)-2 containing 0.5% FBS and incubated with the MFAT conditioned medium for 24 h. The cells were then used for the tube formation assay. The collected media from five samples of the cultured stem cells at two time points (24 h and five days) were used at a 1:2 dilution.

BAEC grown in a complete medium were used as one positive control and cells grown in the EBM2 medium containing the growth factor FGF2 (25 ng/mL) were also used as a second comparative positive control. Cells in a basal EBM2 and a basal Dulbecco’s modified eagle medium (DMEM) were used as a negative control. The cells were incubated in humidified air of 95% with 5% CO_2_ at 37 °C for 24 h. After that, the cells were fixed with 4% PFA for 15 min. The images were obtained using phase contrast microscopy and five fields of view were taken from each of the three wells of a six-well plate where the number of structures was counted. Experiments were repeated at least twice, and a representative example is shown here showing the mean ± SD.

### 2.8. Western Blotting and Measurement of the Phosphorylated-ERK Expression (p-ERK1/2)

Here, the same five samples of conditioned media used in the angiogenesis assay above were applied to BAEC pre-cultured for 48 h in serum poor (0.5% FBS) DMEM again using FGF-2 as a positive control and after 8 min incubation with an MFAT conditioned medium from the 24 h and five day samples at 37 °C. The following protocol was followed.

After rapid washing with ice-cold PBS, cells were lysed with an ice-cold radioimmunoprecipitation (RIPA) buffer (pH 7.5) containing 25 mM Tris-HCl, 150 mM NaCl, 0.5% sodium deoxycholate, 0.5% SDS, 1 mM EDTA, 1 mM sodium orthovanadate (EGTA), 1 mM phenylmethylsulfonyl fluoride (PMSF) and 1% Triton × 100 and 1 μM leupeptin. The protein concentration of cell lysates was determined using the Bradford protein assay (Bio-rad, Watford, UK) and equal quantities of proteins (30 µg) were mixed with a 2 × Laemmli sample buffer boiled in a water bath for 15 min then centrifuged. The samples were separated along with pre-stained molecular weight markers (32,000–200,000 kDa) by 10% SDS-PAGE. Proteins were electro-transferred (Hoefer, Bucks, UK) onto nitrocellulose filters (Whatman, Protran BA85, Germany) and the filters were blocked for 1 h at room temperature in TBS-Tween (pH 7.4) containing 1% BSA. The membrane was then probed with phospho-ERK1/2 (Abcam, Cambridge, UK) diluted in the blocking buffer as indicated and incubated overnight at 4 °C on a rotating shaker. After washing (5 × 10 min in TBS-Tween at room temperature), the membrane was stained with either goat anti-rabbit or rabbit anti-mouse horse radish peroxidase (HRP) conjugated secondary antibodies diluted in TBS-Tween containing 5% de-fatted milk (1:2000, 1 h, room temperature) with continuous mixing. After a further five washes in TBS-Tween, proteins were visualised using enhanced chemi-luminescent detection (ECL, Thermo scientific, Cambridge, UK) and semi-quantitatively identified fold differences compared with house-keeping controls (α-tubulin, Abcam, Cambridge, UK) determined using Image-Lab software. All experiments were repeated at least twice, and a representative example is shown.

### 2.9. The Effects of the MFAT Conditioned Medium on Lipopolysaccharide (LPS) Mediated Macrophage Inflammatory Activity

To induce monocyte differentiation into adherent macrophages, the U937 cells were seeded at an initial density of 0.5 × 106 in 1 mL of growth media with phorbol-12-myristate 13-acetate (PMA) at 50 ng/mL for 72 h in a 12-well plate. Following differentiation, the cells were washed twice with warm Dulbecco’s Phosphate Buffered Saline (DPBS). Cells were treated with LPS 10 ng/mL with or without the MFAT conditioned medium (five individual samples, five days of treatment, 1/10 dilution and pre-incubated 30 min). The supernatant was harvested after 6 h to carry out an ELISA for the measurement of the key pro-inflammatory protein expression of IL-6 and IL-1β (purchased from R&D Systems as a sandwich complete kit).

## 3. Results

The consistent quantification of heterogenous fat in the form of MFAT was firstly important to allow the comparison of samples using various cell and molecular techniques. We established a method to quantify the amounts of MFAT by centrifuging (300 g/5 min) in a PBS solution after 3× washing (removing the interface). We used cut-blue-tips to obtain a 0.25 mL sample and characterised this as one unit.

### 3.1. Analysis of MFAT Cluster Size and CFU Number

The MFAT cluster size represents an important aspect of its therapeutic capability because the maintenance of a perivascular unit is dependent on it remaining as a relatively homogenous viable ‘tissue graft’ in vivo and also the capability for injection requires appropriate sizing. In Figure 1A we have shown a representative microscopic image of the appearance of the MFAT clusters (×40) whilst in Figure 1B we have provided a graph indicating the mean and standard deviation of the cluster size from six patients based on the expected/optimal size (shown as 0.3–0.75 mm (medium)) and those clusters smaller than 0.3 mm (small) or larger than 0.75 mm (large). Whilst the majority of the clusters was within the ‘medium’ size range, there was a significant variation in the total number of these from each individual sample (representing from 51% to 76% of the total number of clusters).

CFU numbers were analysed following the collagenase digestion of the MFAT samples and the culture of the resulting MSCs for a further 10 days. The results showed that CFU-F colonies were clearly formed and visible to the naked eye as shown in Figure 2A. In Figure 2B we have provided a bar graph showing the mean and standard deviation of the numbers of CFU-Fs obtained between the six samples cultured. This shows a significant variation between approximately 50 CFUs and almost 500 CFUs per sample derived from 0.25 mL of MFAT. Figure 2C,D show examples of the magnified Giemsa-stained appearance of the CFU-Fs.

### 3.2. Phenotype of AD-MSCs Obtained from MFAT (One Sample Examination Only)

This experiment was primarily carried out in order to demonstrate the appropriate methodology we used to enable the purification of the MSCs and to confirm they represented ‘stem-like cells’ with the appropriate characteristics hence this was carried out on only one sample. CD90 and CD73 are classic markers of adventitial mesenchymal progenitor cells whilst CD105 indicates that the cells have adipogenic and osteogenic potential and they should be CD45 negative [14]. We have shown here in Figure 3 that 98–100% of all cells were able to be characterised as MSCs, indicating that this was an effective protocol for deriving these stem cells and for the clear distinction from pericytes, endothelial cells and fibroblastic cells for example.

### 3.3. Characterisation of Cytokine Secretion from MFAT Samples

Cytokines and growth factors present in the conditioned medium from MFAT cultures (in a serum free culture after one day and five days) were measured using our specifically designed multiplex quantitative system representing key markers of inflammation, regenerative responses, anti-bacterial and vasculogenesis.

Figure 4A shows an example of the heat map of the 32 factors after a serum free culture of MFAT from one of our eight healthy donors after 24 h culture. Most interestingly, there was a notable production of the IL-1Rα antagonist with a significant increase between one day and five-day culturing in serum free MEMα (Figure 4B; *p* < 0.05). Figure 4C shows a significant production of CXCL9 also increasing between one day and five-day culturing in serum free MEMα (*p* < 0.05). In Figure 4D, a slightly different profile for hepatocyte derived growth factor (HGF) was found with notable quantities of HGF produced even after 24 h and a maintenance of secretion over five days rather than a further significant increase (*p* > 0.05) (Figure 4D); macrophage migration inhibitory factor (MIF) protein was also produced in significant quantities after 24 h culture of MFAT and this became significantly higher after five days of culture (*p* < 0.05) (Figure 4E). Both TGFβ1 and β3 were produced and secreted into the medium by MFAT and levels significantly increased over the five-day incubation period Figure 4F–G). Appendix A shows the raw data from the cytokine analysis. Overall, these data suggest a concerted or increasing secretion of critical cytokines over time by MFAT even when cultured as here in non-stimulating (SFM) conditions.

### 3.4. Effects of the Conditioned Medium from MFAT on Angiogenesis and Associated Cell Signalling

A tube-like structure formation was assessed in BAECs exposed to the MFAT conditioned medium and the results showed that an increase in angiogenic structures were seen in 3/5 using the 24 h conditioned medium and 5/5 from the five day conditioned medium compared with the negative control basal medium (Figure 5A and graph in Figure 5B). Note the difference between samples in the extent of the stimulation of the tube-like structure formation. P-ERK1/2 is a key signalling molecule known to increase during the process of angiogenesis and is associated with migration and proliferation. We have shown here that the same samples caused a notable increase in the expression of p-ERK1/2 with once again a distinct variation from sample to sample. Β-tubulin was used as a house-keeping control and the bar graph represents the relative changes in ‘fold’ compared with the control but was not quantitative.

### 3.5. The Effects of the MFAT Conditioned Medium on the LPS-Stimulated Macrophage Expression of Pro-Inflammatory Cytokines

Differentiated U937 monocytes responded to LPS inclusion in the medium by producing significant quantities of the cytokines Il-1β and Il-6 as shown in Figure 6A,B, respectively. The release of both cytokines was completely attenuated in the presence of MFAT from all samples tested demonstrating a strong anti-inflammatory capacity.

## 4. Discussion

The data presented here for the first time identified specific classes of cytokines produced/released in notable quantities over time from a collection of MFAT preparations providing an explanation of the capacity of this injectable material to modulate inflammation, pain, tissue repair and possibly regeneration in a variety of conditions from osteoarthritis (O/A) to diabetic wound healing. The series of experiments chosen here was designed to provide a clear series of data that would fully characterise the potency of samples, identify critical differences in their potential biological activity or capacity to function as effective tissue grafts and provide some evidence to qualify the clinical outcome data that suggested only an approximate 80% positive response rate to treatment with 10% super responders thereby eventually providing a framework to help to stratify patient treatment choice with MFAT.

The significant variation in the amounts of release from sample to sample (person to person) and the variability in the cluster size and CFU-F producing capability of the isolated cells provided a possible insight into the reason for the differences in effectiveness when given as a treatment between individuals. From this we could extrapolate the possibility that the lowest activity samples might be less effective therapeutically indicating a need to characterise and refine the technology to optimise and maximise treatment benefits.

Our data describing MFAT cluster size obtained from the Lipogems^®^ International Spa micro-fragmentation system [10] and CFU production capability following SVF purification suggested that there was a relatively homogenous preparation of mid-range sized tissue fragments that maintained their original architecture and perivascular stromal components and that harvested MSCs maintained their capacity to form CFUs that matured and expanded over time. Nonetheless, there is room for optimisation because smaller clusters tend to perish whilst larger ones have a relatively lower surface area and potentially lower activity. This would then be magnified when we see the variation in the capability to produce CFUs, which provided the strongest indication of a capability to secrete cytokines and growth factors [15].

Several previous studies have looked at the expression profiles of cytokines released by MFAT and SVF in a culture [6] although only using available commercial arrays and showed the MFAT expression of angiogenic factors such as the placental growth factor and angiogenin as well as HGF (shown in our data here also) and PDGF growth factors and other inflammation associated molecules such as IL-13, 3, 16 and 27. Their purpose was to compare the secretion against that of SVF and whilst MFAT was clearly more active in its secretion profile the comparisons failed to include key markers that defined the capability of MFAT; for example, to reduce pain and support the repair of or regeneration in joints with osteoarthritis [16]. Here, we have constructed an array that identified the major players, demonstrating a very high expression of the critical anti-inflammatory protein IL1-Rα antagonist, the major ‘protector’ against Il-1 signalling, which is the critical inflammatory pathway activated during the development of O/A and blocking, which has been shown to nullify multiple connected inflammatory pathways in disease [17]. Our data in addition showed a varied ability between MFAT samples (conditioned media) to stimulate angiogenesis and MAP kinase signalling activation providing further evidence of a possible difference in the effectiveness from patient to patient in respect of a response to treatment. Of note was the contradiction in the higher increased p-ERK expression from the day one conditioned medium versus higher tube-like structure formation in the day five conditioned medium. There may be other cytokines/growth factors that in combination account for these differences and these should be examined in further detail.

CXCL-9, also identified here, has a critical role in innate paracrine signalling and M2 macrophage phenotypic resolution associated with anti-inflammatory and immunosuppressive functions as well as the induction of vascular stability [18]. In addition, this cytokine has a strong direct anti-microbial activity providing a protective mechanism against potential infection during and after treatment [19]. Similarly, the macrophage migration inhibitory factor (MIF) has recently been shown to be associated with protection against O/A [20] and may also be important in modulating autoimmune reactions such as those associated with rheumatoid arthritis, lupus or giant cell arthritis. Of note, Bougle et al. [8] have already demonstrated the ability of MFAT samples to protect against the systemic development of inflammation in vivo using an animal model of caecal ligation.

Regarding wound healing and regenerative capacity, as already mentioned, HGF is an important molecule for the co-ordination of reformation of perivascular and other tissue architecture [21], whilst our novel finding of the significant output of TGFβ1 and β3 by MFAT with known functions as both immune suppressors and key stimulators of scarless and fibrosis free wound healing [22] indicated a group of proteins described in this work that together provide a potentially highly enriched micro-environment associated with MFAT tissue graft ideally suited for long-term anti-inflammatory pain relief and for the support of damage repair not only in O/A but potentially in other tissue and organ recovery.

Future perspectives suggest that MFAT, being a long-lasting, persistent and ‘sticky’ tissue graft, has the biological characteristics defined here that make it highly suitable as a conservatory and protective treatment against pain and worsening of O/A. This is the first study to effectively characterise the major critical cytokines, classify the cluster and CFU capabilities and demonstrate the inherent significant variability in potency from the material obtained from patient to patient thereby providing evidence to support the clinical findings where there is a difference in response in patients with a seemingly similar O/A grade and condition. This is important for clinics to be able to determine the most likely form of useful treatment and in future more elaborate clinical studies/trials should aim to include this type of data to create an artificial intelligence platform that will inform clinicians of the most effective treatment pathway for individuals.

Study limitation: We have to acknowledge that our data on the release of cytokines from MFAT were based on secretion in a neutral serum free culture environment and that this may not reflect their activity and secretion in vivo.

## Figures and Tables

**Figure 1 ijms-22-03271-f001:**
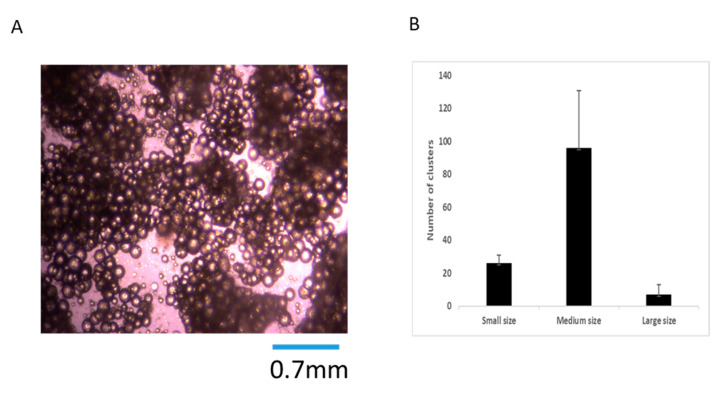
Micro-fragmented adipose tissue (MFAT) indicated most of the clusters were medium sized. (**A**); MFAT image of clusters seen under the microscope (×40). (**B**); The number of medium sized clusters (0.3–0.75 mm in diameter) from MFAT of six patients was significantly higher than the smaller sized clusters (<0.3 mm) and large sized clusters (>0.75 mm) (*p* < 0.01) in a 10 field measured at ×40 magnification per sample.

**Figure 2 ijms-22-03271-f002:**
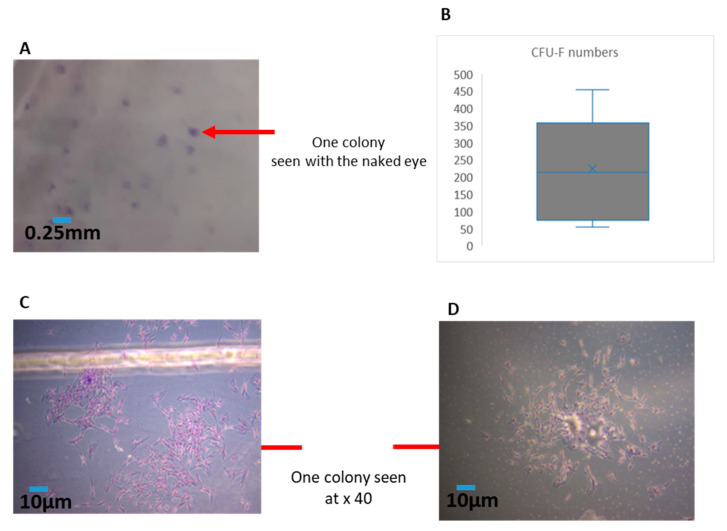
The numbers of fibroblastoid colony-forming unit (CFU-F) colonies of 0.25 mL of MFAT from the six patients in Figure 1 indicated a large variation. A total of 0.25 mL MFAT was digested with collagenase in MEMα and then cultured in a MEMα complete medium for 10 days, stained with Giemsa and fixed with 100% methanol. (**A**); shows colonies observed with the naked eye. (**B**); the mean numbers of colonies ± SD. (**C**,**D**) show the magnified appearance of an individual colony under microscopy (×40).

**Figure 3 ijms-22-03271-f003:**
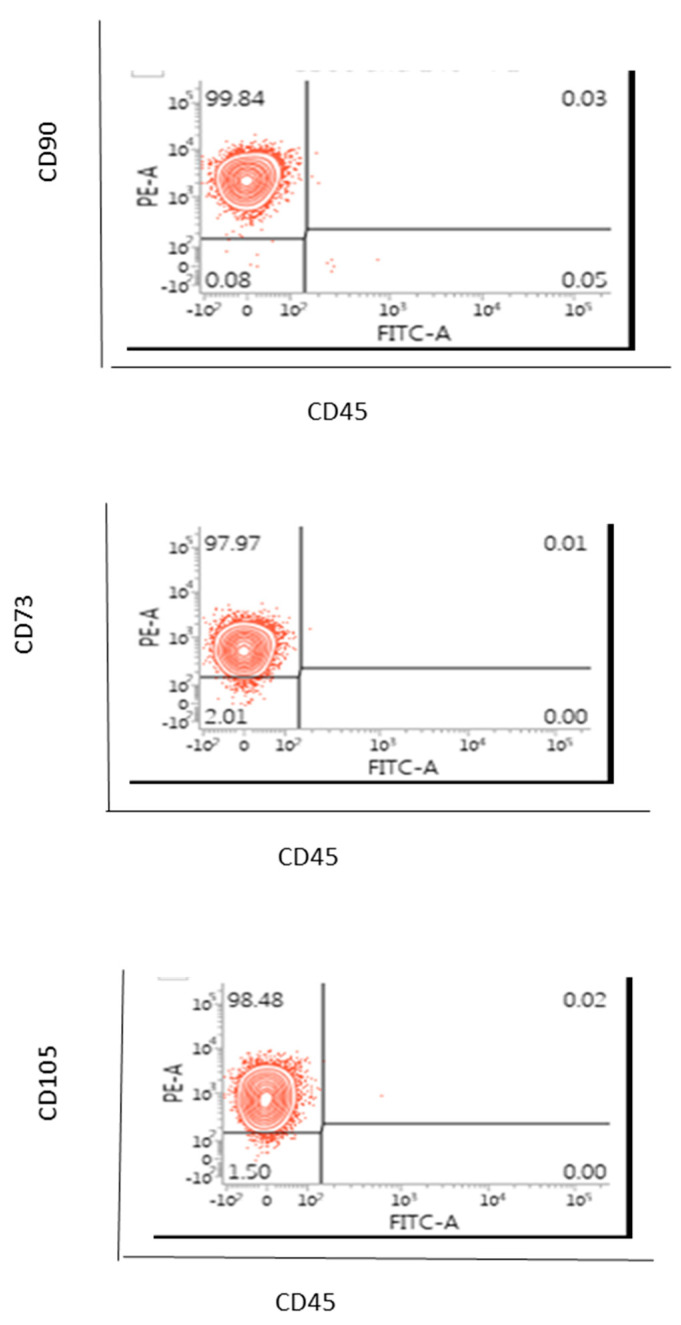
Phenotype of passage 4 of AD-MSCs from patients’ MFAT. Almost all of the cells (>98%) indicated CD90+CD45-, CD73+CD45-, CD105+CD45- showing stem cell/ mesenchymal stem cell (MSC) origin and capacity. One sample only shown here.

**Figure 4 ijms-22-03271-f004:**
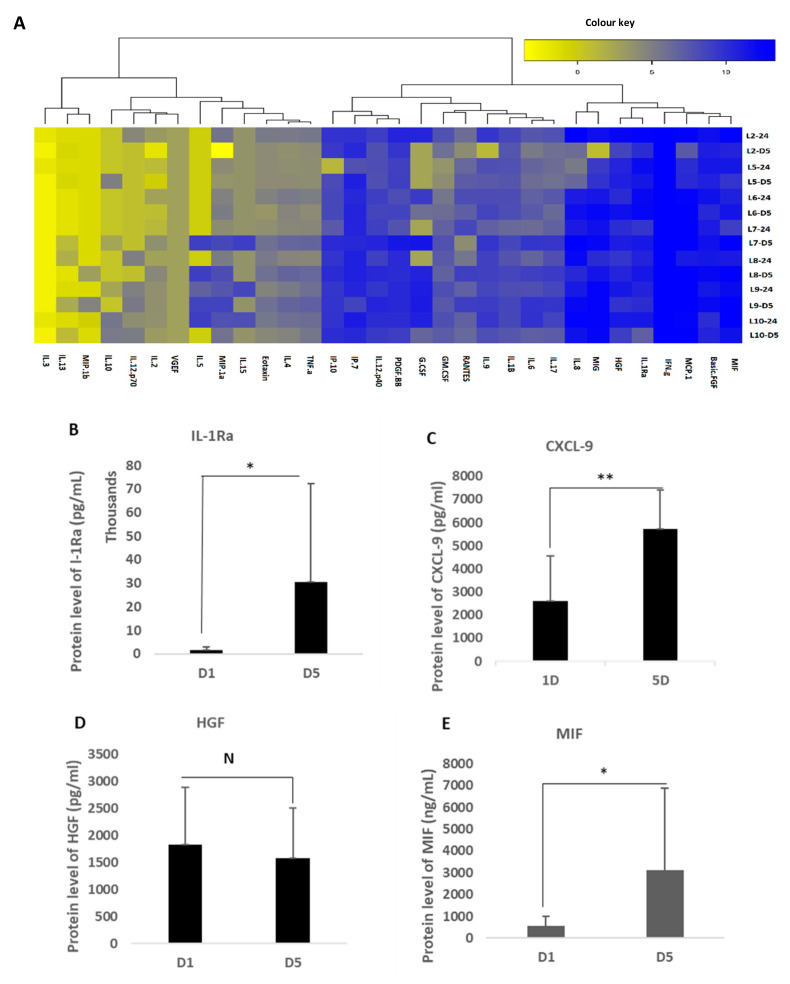
Cytokines and growth factors in the MFAT condition medium in a serum free culture for one day and five days. (**A**)**;** shows an example of a heat map of 32 factors after a serum free culture of MFAT (carried out on 7/8 of the donors). With multiplex analysis, 50 µL of the MFAT serum free culture condition medium was detected for 32 cytokines and growth factors. Other heat maps and raw data are shown in Appendix A. (**B**); There was a significant increase in IL-1Ra between one day and five-day culturing in serum free MEMα (*p* < 0.05 *). (**C**); There was a significant increase in CXCL9 between one day and five-day culturing in serum free MEMα (*p* < 0.05 *). (**D**); The expression of HGF was found but this remained stable between one day and five-day culturing (*p* > 0.05 *). (**E**); MIF indicated significantly higher in five-day culturing than in one day cultures (*p* < 0.05). (**F**–**G**); TGFβ1-3 secretion significantly increased in the MFAT serum free culture condition medium on day five (<0.01 **). Similarly, TGFβ3 secretion significantly increased in the MFAT serum free culture condition medium on day five as well (<0.01 **).

**Figure 5 ijms-22-03271-f005:**
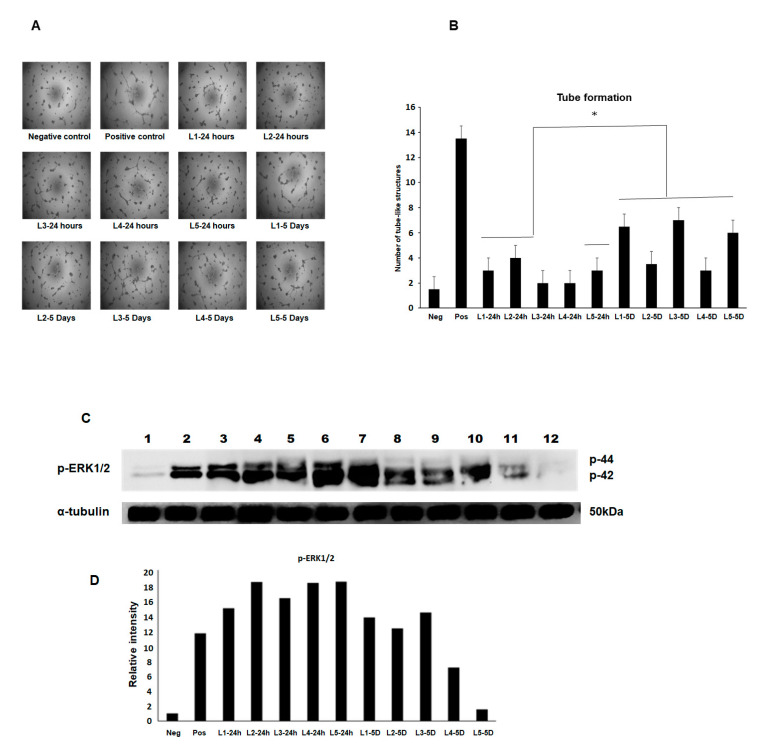
The pro-angiogenic effect of the MFAT serum free culture condition medium collected at day one and day five. The day five conditioned medium indicated a higher pro-angiogenic effect on the production of tube-like structures in BAEC than day one (*p* < 0.05) (**A**,**B**). Western blotting in (**C**,**D**) shows the increased notable relative p-ERK expression in the conditioned medium of MFAT cultures after 24 h with a reduction in the day five conditioned medium. Lanes: (individual samples labelled as L) 1: Negative control (DEM, basal medium), 2: Positive control (EBM2, completed medium), 3: L1 24 h, 4: L2 24 h, 5: L3 24 h, 6: L4 24 h, 7: L5 24 h, 8: L1 five days, 9: L2 five days, 10: L3 five days, 11: L4 five days, 12: L5 five days.

**Figure 6 ijms-22-03271-f006:**
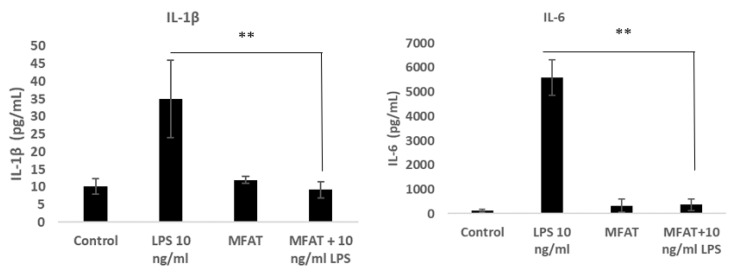
The inhibitory effect of the MFAT condition medium on the LPS mediated cytokines secretion MFAT conditioned medium indicated a significant inhibition on LPS (10 ng/mL) mediated IL-1ß1 secretion (*p* < 0.01 **). (**A**); the MFAT condition medium indicated a significant inhibition of LPS (10 ng/mL) mediated IL-1β (*p* < 0.01 **). (**B**); Il-6 expression. n = 5 samples tested from the day five conditioned medium of MFAT cultures by an ELISA.

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
