# Peer review of "Characterisation of Novel Angiogenic and Potent Anti-Inflammatory Effects of Micro-Fragmented Adipose Tissue"

_ijms, 2021, doi:10.3390/ijms22063271_

Round 1

Reviewer 1 Report

The MS is about the characterization of novel angiogenic and potent anti-inflammatory effects of micro-fragmented adipose tissue. This work is well organized and written. Also, the content looks novel.

It can be accepted.

Author Response

Many thanks for the constructive and kind comments of the reviewer we are delighted you have considered this appropriate for publication in IJMS

BW

Mark

Reviewer 2 Report

This work is potentially interesting, but depends on the other previous works too much. Each results in this paper are not connected, seem to be superficial. The results are not the strong direct evidence why MFAT can have angiogenic and anti-inflammatory effect. May be because the authors used the conditioned medium to analyze this. Will the changed level of the cytokines affect also in vivo? If MSCs are the main cell types of MFAT effect, MSC clusters may be used instead of MFAT for transplantation to O/A. The authors have to declare the merits using MFAT.  Moreover, there are a lot of misspells and insufficient presentations (see the attached file).

In page 11, Table 2 is stated, but there is no Table 2.

The authors mentioned TGF-beta1 and beta3 in the Figure 4 results, but that kind of data was not shown in anywhere.

In Figure 5, 5 days conditioned medium is more effective in tube formation than 1 day. However, 1 day conditioned medium has more pERK than 5 days. This is contradictory. 

Author Response

We thank the reviewer for their comments and have incorporated changes to the manuscript with these in mind. Specifically,

This work is potentially interesting, but depends on the other previous works too much. We have based the work on our use of Lipogems derived MFAT over the last few years and the varied clinical response of patients treated for knee O/A. We have been collaborators of the authors of the 2 works that so far have made some assessment of the MFAT characteristics in the literature those being Vezzani et al and Nava et al and we would like to clarify that based on their findings we realised they were not able to effectively characterise the major critical cytokines nor understand why potentially there is a difference in response in patients from MFAT sample to MFAT sample. This is important for the clinics to be able to determine the most likely form of useful treatment-we have added an explanatory paragraph at the end of the discussion justifying this and going on to define what the future clinical study should look like as part of an AI stratifying programme. 

Each results in this paper are not connected, seem to be superficial. This rationale has also been clarified at the start of the discussion section being based on setting up a series of biologically clarifying methods that can be used to characterize and stratify the potency of any particular sample for use clinically in O/A treatment 

The results are not the strong direct evidence why MFAT can have angiogenic and anti-inflammatory effect. Please see the statement in the discussion of the importance of IL-1R alpha antagonist in blocking inflammation per se and its major role in O/A we have elaborated also on this and also a statement about the results of the angiogenesis experiments.

May be because the authors used the conditioned medium to analyze this.

Will the changed level of the cytokines affect also in vivo? If MSCs are the main cell types of MFAT effect, MSC clusters may be used instead of MFAT for transplantation to O/A. The idea of using MFAT comes from the importance of using a 'sticky' fat graft in part created by the presence of adipocytes in the clusters-that ourselves and others have shown is able to remain viable for months and years at the site of injection. Further more as it retains the architecture of the original fat it is classified as minimally manipulated autologous treatment whilst CFUs of stem cells would be classified as a novel therapeutic and currently are not licensed for use. We hope this explanation is sufficient for the reviewer? Our collaborators published recently a paper showing that in vivo MFAT injection protects against caecal induced sepsis providing in vivo evidence that the MFAT operates in the same way. we have included this in the discussion.

The authors have to declare the merits using MFAT.  Moreover, there are a lot of misspells and insufficient presentations These have been corrected and the importance of MFAT added to in the discussion

Table 2 should have been included we will check with the editors thank you for highlighting this

Similarly figure 4 TGF results images seem to have bee missed by the editor we will ensure they are included

It is interesting to see that the p-ERK data shows higher activity in the 24h conditioned media samples-although WB is only semi-quantitative at best, still, this may suggest a difference in expression of specific growth factors or cytokines between day 1 and day 5 and this is something that requires further study to identify -the contradiction looks true although the tube-stimulation was not particularly strong even at 5 days this could be due to un tested factor differences such as in expression of angiopoietin-2 and needs to be studied further a statement on this has been added

Round 2

Reviewer 2 Report

The manuscript is improved, but there are a few comments the authors did not respond.

Still Table 2 (Line 296) is missing. Do authors mean Supplemental Table 2?

I hope the authors would add a scale bar in Figure 2 because readers know the size more easily.

This reviewer understands the importance of MFAT analysis via response from the authors. However, still it is not clear cytokine level measured in the conditioned medium at certain time points reflects in vivo cytokine changes with time. The environment of in vivo and the culture should be very different. The authors have to address this issue. Or, have to declare as a limitation.

Author Response

We thank the reviewer for their diligence and comments:

below are the responses:

The scale bar has been added to the figure 2A apologies for missing this

Table 2 has been corrected to mean supplementary-both Tables are supplementary

we have added the following

Study limitation: We have to acknowledge that our data on the release of cytokines from the MFAT are based on secretion in a neutral serum free culture environment and that this may nor reflect their activity and secretion in vivo. 

Round 3

Reviewer 2 Report

The manuscript has been improved. The authors should add the scale bar in all pictures of Figure 2. 

Author Response

I have added the scale bars to figure 2 C and D

Many thanks indeed